# Antioxidative Effects of Curcumin on the Hepatotoxicity Induced by Ochratoxin A in Rats

**DOI:** 10.3390/antiox10010125

**Published:** 2021-01-17

**Authors:** Sara Damiano, Consiglia Longobardi, Emanuela Andretta, Francesco Prisco, Giuseppe Piegari, Caterina Squillacioti, Serena Montagnaro, Francesco Pagnini, Paola Badino, Salvatore Florio, Roberto Ciarcia

**Affiliations:** 1Department of Veterinary Medicine and Animal Productions, University of Naples “Federico II”, Via Federico Delpino n.1, 80137 Naples, Italy; emauela.andretta@unina.it (E.A.); francesco.prisco@unina.it (F.P.); giuseppe.piegari@unina.it (G.P.); caterina.squillacioti@unina.it (C.S.); serena.montagnaro@unina.it (S.M.); florio@unina.it (S.F.); 2Department of Mental, Physical Health and Preventive Medicine, University of Campania “Luigi Vanvitelli”, Largo Madonna delle Grazie n.1, 80138 Naples, Italy; consiglia.longobardi@unicampania.it; 3Unit of Radiology, Department of Medicine and Surgery, University of Parma, Via Gramsci 14, 43126 Parma, Italy; f.pagnini90@gmail.com; 4Department of Veterinary Science, University of Turin, L. go P. Braccini 2-5, 10095 Grugliasco, Italy; paola.badino@unito.it

**Keywords:** ochratoxin A, curcumin, oxidative stress, liver, toxicity

## Abstract

Ochratoxin A (OTA) is a powerful mycotoxin found in various foods and feedstuff, responsible for subchronic and chronic toxicity, such as nephrotoxicity, hepatotoxicity, teratogenicity, and immunotoxicity to both humans and several animal species. The severity of the liver damage caused depends on both dose and duration of exposure. Several studies have suggested that oxidative stress might contribute to increasing the hepatotoxicity of OTA, and several antioxidants, including curcumin (CURC), have been tested to counteract the toxic hepatic action of OTA in various classes of animals. Therefore, the present study was designed to evaluate the protective effect of CURC, a bioactive compound with different therapeutic properties on hepatic injuries caused by OTA in rat animal models. CURC effects were examined in Sprague Dawley rats treated with CURC (100 mg/kg), alone or in combination with OTA (0.5 mg/kg), by gavage daily for 14 days. At the end of the experiment, rats treated with OTA showed alterations in biochemical parameters and oxidative stress in the liver. CURC dosing significantly attenuated oxidative stress and lipid peroxidation versus the OTA group. Furthermore, liver histological tests showed that CURC reduced the multifocal lymphoplasmacellular hepatitis, the periportal fibrosis, and the necrosis observed in the OTA group. This study provides evidence that CURC can preserve OTA-induced oxidative damage in the liver of rats.

## 1. Introduction

Oxidative stress has become a new hot spot in the context of mycotoxin mechanism of action [1]. In fact, oxidative stress is hypothesized to be one of the main causes in the development of many disorders such as chronic kidney disease, hepatic inflammation, hypercholesterolemia, diabetes, and hepatic cirrhosis [2]. Oxidative stress involves an excessive production of free radicals, which, in turn, induce oxidative damage to cellular biomolecules, including proteins, lipids, and nucleic acids, in numerous tissues [3]. Several studies indicate that oxidative stress plays critical roles in the toxicity of ochratoxin A (OTA) [4,5]. To date, more than 300 mycotoxins have been identified; however, due to their toxic levels and contamination profiles, several groups are particularly interesting, including aflatoxins, OTA, trichothecenes (including T-2 toxin, deoxynivalenol), fumonisins, zearalenone, and patulin [6,7]. OTA, a mycotoxin produced by *Aspergillus* and *Penicillium* genera, is one of the most detected mycotoxins in food and foodstuffs such as coffee beans, fermented tea, and cereals [8]. This toxin inflicts losses to farmers and reduces the value of contaminated feeds. Effects in animals subjected to the ingestion of these fungal compounds vary from acute, overt disease with high morbidity and death to chronic, decreased resistance to pathogens and reduced animal productivity [9,10]. The major clinical complication associated with animal feed contaminated with mycotoxins is not acute disease, but rather chronic disease, caused by the ingestion of small quantities of poisoned food which may lead to an array of metabolic, physiologic, and immunologic disorders [9,10]. OTA often causes chronic toxicity, due to the prolonged intake of its small amounts, and manifests itself with hepatotoxic, carcinogenic, genotoxic, teratogenic, nephrotoxic, and immunosuppressive effects, affecting both humans and numerous animal species [11,12]. Among the animal productions, the risk is limited to monogastric species since the OTA amide bond can be hydrolyzed by ruminants and form a nontoxic molecule [13]. Pigs are the most susceptible to the accumulation of OTA, whose tissue deposition occurs as follows: kidney > liver > muscle > fat [14]. In addition, OTA was classified by the International Agency for Research on Cancer (IARC) in group 2B, as possibly carcinogenic to humans, based on the kidney and liver tumors reported in mice and rats [15]. The main organ subjected to toxicological studies is the liver, since this organ has a special role in the metabolism, storage, redistribution, and excretion of endogenous and exogenous substances in the body [16]. Even if OTA effects are known, the molecular mechanisms underlying the damage are still not completely clarified. OTA exposure (in vitro or in vivo) has been related to overproduction of reactive oxygen species (ROS), as well as oxidative damage (lipids, proteins, and DNA). In addition, OTA may reduce the antioxidant defense of cells by reducing GSH and cytoprotective enzymes [17]. It has been demonstrated that antioxidants could protect cells against OTA-induced cytotoxicity and genotoxicity [18,19]. The use of bioactive compounds has emerged as a potential approach to reduce toxicity induced by environmental contaminants such as mycotoxicosis. Curcumin (CURC), a polyphenolic compound, is a natural bioactive constituent isolated from the rhizome of *Curcuma longa* Linn. Several studies have reported that CURC has numerous pharmacological activities, including antioxidant, anti-inflammatory, antitumor and anti-bacterial effects [20]. CURC has strong antioxidant activity by exerting its effect on reactive species, scavenging superoxide anion (O^−^), peroxynitrite (NOO), nitric oxide (NO), peroxyl radicals (ROO), and hydroxyl (OH^−^) radicals, resulting in the upregulation of antioxidant proteins [21]. Phenolic groups of CURC are responsible for its ability to react with reactive species and might likely be one of the mechanisms via which CURC administration protects cells from oxidative damage. In fact, CURC can indirectly induce the expression of antioxidant proteins such as superoxide dismutase (SOD), catalase (CAT), glutathione peroxidase (GPx), glutathione reductase (GR), glutathione-S-transferase (GST), and g-glutamyl cysteine ligase (gGCL) [21]. The hepatoprotective effects of CURC against toxic chemical-induced liver injury have already been explored and have been attributed to its intrinsic antioxidant properties [20,22,23,24]. Moreover, recently, S.S. Zhai and colleagues have shown that dietary supplementation of CURC reversed serum biochemical changes and ameliorated liver oxidative injury in White Pekin ducklings treated for three weeks with OTA [25]. 

However, until now, the OTA response to differences in species was far from clear [26]. Various in vivo and in vitro studies have identified several diverse metabolites of OTA in different species that could be the cause of slight differences from animals [27,28,29]. Therefore, the present study has been designed to investigate the efficacy of CURC on OTA-induced hepatotoxicity. 

## 2. Materials and Methods

### 2.1. Chemicals

OTA and CURC were supplied by Sigma-Aldrich (Milan, Italy). SOD (Item No. 19160), malondialdehyde (MDA) (Item No. MAK085), GPx (Item No. 38185), and CAT (Item No. CAT100) assay kits were purchased from Sigma-Aldrich (Milan, Italy). Other chemicals and reagents used in this work were purchased from Sigma-Aldrich (Milan, Italy). The animal supplier was Charles River Laboratories (Milan, Italy).

### 2.2. Ethics Statement

The use and care of the animals in this work was approved by the Institutional Animal Care and Ethics Committee (Approval Number: 487/2018-PR) and carried out in accordance with the associated guidelines EU Directive 2010/63/EU.

### 2.3. Experimental Design and Sample Collection

Twenty-four male rats of the Sprague Dawley strain, 10 weeks old (250–270 g), used in this study were randomly distributed into four experimental groups (6 rats for each group) and were housed in cages under standard conditions (temperature 20 ± 2 °C and 12 h day/night cycles). The animals received a standard diet ad libitum. Animals were treated daily for 14 days by gavage as follows: CONTROL group: 2 mL/kg b.w. of olive oil; OTA group: 2 mL/kg b.w. of olive oil containing 0.5 mg/kg b.w. of ochratoxin A [30]; CURC group: 2 mL/kg b.w. of olive oil containing 100 mg/kg b.w. of curcumin [31]; OTA (2 mL/kg b.w. of olive oil containing 0.5 mg/kg b.w. of ochratoxin A) + CURC (1 mL/kg b.w. of olive oil containing 100 mg/kg b.w. of curcumin). The use of olive oil has served to improve the stability of CURC. The duration of the experiment (14 days) was based on our previous work [30,32,33]. At the end of the experimental period, rats were anesthetized with 2% isoflurane (Isotec 4, Palermo, Italy), and after complete sedation, blood samples were collected from the aorta into nonheparinized bottles and processed to aliquots for biochemical analysis. At the end of the treatment, rats were sacrificed by cervical dislocation, and the kidney and liver were removed to measure the oxidative stress markers and lipid peroxidation and were partially prepared for routine histopathology. Kidney results are reported in our previous paper [34] where CURC has shown a good recovery of kidney damage induced by OTA.

### 2.4. Determination of Serum Hepatic Function Biomarkers

The activities of the hepatic function biomarkers alanine aminotransferase (ALT), aspartate aminotransferase (AST), and alkaline phosphatase (ALP) were measured after 14 days of treatment by an auto-chemistry analyzer (PKL PPC 125, Paramedical srl, Salerno, Italy) following the instructions of the manufacturer of the commercial diagnostic kits, expressing data in units per liter (U/L). Total protein concentrations of the serum were colorimetrically determined [35].

### 2.5. Determination of Liver Antioxidant Enzyme Activities and Malondialdehyde

Liver samples from all groups were collected on day 14 of the treatment. One gram of each liver sample was added to 9 mL of normal saline 0.9% and homogenized in ice, using electrical tissue homogenizer (Tissue Lyser, Qiagen, Milano, Italy), and centrifuged at 10,000× *g* for 15 min at +4 °C; resulting supernatants were stored at −80 °C. Then, the supernatant was used to evaluate, by a spectrophotometer (Glomax Multi detection system, Promega, Milano, Italy), the SOD, CAT, and GPx activities according to previous studies [36,37,38]. These activities were expressed as units per milligram of protein (U/mg of protein). Malondialdehyde (MDA), a marker of lipid peroxidation, was calculated according to Ohkawa et al. [39]. The optical density (OD) of the supernatants was read by a spectrophotometer at a wavelength of 532 nm and was expressed in nanomoles of MDA per milligram of protein. 

### 2.6. Histopathological Studies 

Livers of 24 male Sprague Dawley rats (6 rats per group), collected during necropsy, were fixed in Bouin solution for 24 h and subsequently dehydrated in ascending ethyl alcohol and then embedded in paraffin. Two serial sections at 3 μm were stained with hematoxylin and eosin and with Masson’s trichrome stain and were examined and photographed with a light microscope (Nikon Eclipse E600) coupled with a microphotography system (Nikon digital camera DMX1200). Hepatic lesions were scored by evaluating at least 10 microscopic fields at 20× magnification and using already defined scoring systems. Notably, inflammation was scored as follows: score 0, no inflammatory foci; score 1 (mild), <2 foci per 20× field; score 2 (moderate), 2–4 foci per 20× field; score 3 (severe), >4 foci per 20× field. The extent of the steatosis was scored as follows: score 0, <5% of hepatocytes; score 1 (mild), 5–33%; score 2 (moderate), >33–66%, score 3 (severe), >66% [40]. The extent of the necrosis was scored as follows: score 0, 0% of hepatic tissue; score 1 (mild), <10%; score 2 (moderate), >10–50%, score 3 (severe), >50% [41]. Furthermore, the presence or absence of fibrosis, sinusoidal dilation, and central vein dilation was recorded for each case [42].

### 2.7. Statistical Analysis

Statistical analysis of enzymatic activities was expressed as mean ± standard deviation (SD). Analysis of variance (ANOVA) tests followed by a Tukey’s test were used to analyze the differences (GraphPad Software 3.00, San Diego, CA, USA). Each animal group consisted of 6 rats and the experiment was conducted in triplicate. Values of * *p* < 0.05 were considered statistically significant.

Statistical analysis of the liver histology was performed using IBM SPSS Statistics (Version 25) with a level of significance of 0.05. Each animal group consisted of 6 rats and the experiment was conducted in triplicate. The differences in the distribution of the histologic semiquantitative scores among groups were compared using the Kruskal-Wallis H test and a post hoc multiple comparison using Dunn’s test. The difference in frequency of fibrosis, sinusoidal dilation, and central vein dilation among groups was evaluated with a two-tailed Fisher’s exact test.

## 3. Results

### 3.1. Effect of CURC on Liver Biochemical Analyses in Rats

The activities of the hepatic function biomarkers ALT, AST, and ALP and the total protein in rats treated after 14 days of treatment are presented in Table 1. The oral administration of OTA caused a significant increase in ALT, AST, and ALP activities (increases of −55.7%, −37.1%, and −71%, respectively) and caused a significant reduction in the concentration of total protein (reduction of 53.1%) compared to the CONTROL group after 14 days of treatment. However, the cotreatment with CURC significantly reduced the adverse effects of OTA at the end of treatment, significantly lowering ALT, AST, and ALP activities by 30.7%, 19.8%, and 54.1%, respectively, and increasing the concentration of the proteins in the serum by −31%.

### 3.2. Activity of Antioxidant Enzymes SOD, CAT, and GPx

The antioxidant markers SOD, CAT, and GPx in the livers of the rats in the different experimental groups after 14 days of treatment are shown in Figure 1a–c. The activities of SOD, CAT, and GPx were significantly decreased in the liver of OTA-treated rats in comparison to the CONTROL group. In fact, SOD value was 7.51 ± 0.5 in CONTROL group and 6.30 ± 0.3 in OTA group (* *p* < 0.05), CAT value was 2.66 ± 0.1 in CONTROL group and 2.23 ± 0.2 in OTA group (* *p* < 0.05), and GPx value was 13.5 ± 1.2 in CONTROL group and 5.1 ± 1.5 in OTA group (**** *p* < 0.0001). Cotreatment with CURC showed a good recovery of SOD, CAT, and GPx activities compared to the OTA group. In fact, the SOD value was 8.25 ± 0.4 in the OTA plus CURC group, compared to 6.30 ± 0.3 in the OTA group (^#^
*p* < 0.05). The CAT value was 2.95 ± 0.1 in the OTA plus CURC group, compared to 2.23 ± 0.2 in the OTA group (^#^
*p* < 0.05). The GPx value was 10.2 ± 2.1 in the OTA plus CURC group, compared to 5.1 ± 1.5 in the OTA group (^####^
*p* < 0.0001). When CURC was used alone, no change in SOD, CAT, and GPx activities was observed when compared to the CONTROL group. In fact, the SOD value was 8.1 ± 0.8 in the CURC group, compared to 7.51 ± 0.5 in the CONTROL group. The CAT value was 2.53 ± 0.14 in the CURC group, compared to 2.66 ± 0.1 in the CONTROL group. The GPx value was 15.4 ± 2.6 in the CURC group, compared to 13.5 ± 1.2 in the CONTROL group.

### 3.3. Lipid Peroxidation

MDA levels in the liver tissues were significantly increased in OTA when compared to CONTROL group (0.38 ± 0.024 in CONTROL compared to 0.90 ± 0.031 in OTA (**** *p* < 0.001)). Cotreatment with curcumin showed a significant decrease in MDA concentration compared to the OTA group. Indeed, the MDA value changed from 0.99 ± 0.031 (OTA) to 0.42 ± 0.015 (OTA+ CURC) (^#^^#^^##^
*p* < 0.001). When CURC was used alone, there was no change in MDA levels compared to the CONTROL group (0.35 ± 0.012 in CURC compared to 0.38 ± 0.024 in CONTROL) (Figure 2).

### 3.4. Histopathological Examination

Rat livers from CONTROL and CURC-treated groups rarely showed mild, multifocal, random lymphoplasmacytic inflammation, and mild to severe steatosis. No necrosis was observed in the livers of rats in CONTROL and CURC-treated groups. No statistical differences were observed between CONTROL and CURC treated groups regarding inflammation, steatosis, necrosis, and fibrosis. Central vein dilation was more frequent in the CURC group compared to the CONTROL group. Differently, livers from the OTA-treated group showed moderate to severe, multifocal, random lymphoplasmacytic inflammation, moderate to severe steatosis, and moderate to severe necrosis. Portal spaces were often multifocally expanded by moderate to severe fibrosis. OTA group showed a more severe inflammation compared with CURC group (*p* < 0.05) and a more severe necrosis compared with both CONTROL (*p* < 0.01) and CURC groups (*p* < 0.01). Fibrosis was more frequently observed in the OTA group when compared with both CONTROL and CURC groups (*p* < 0.05). Central vein dilation was observed more frequently in OTA-treated rats compared with CONTROL (*p* < 0.05). No significant differences were observed in inflammation, steatosis, necrosis, fibrosis, and central vein dilation comparing OTA + CURC with CONTROL, CURC, and OTA groups. No significant differences in sinusoidal dilation frequency were observed among groups (Figure 3A,B).

## 4. Discussion

OTA is a widely spread worldwide mycotoxin and represents an emerging health problem for both humans and animals, due to its deleterious effects and its high presence in feed and food [43]. In consequence, the European Union (EU) and Joint Food and Agriculture Organization (FAO)/World Health Organization (WHO) have evaluated the risk assessment of OTA and recommended regulatory levels to control or prevent OTA contamination [44]. Therefore, OTA has received utmost special attention from professionals of microbiology, toxicology, and food technology. Despite several in vivo and in vitro studies published on the nephrotoxicity and hepatotoxicity of OTA, the exact mechanisms involved, as well as the influence of oxidative stress on the deleterious effects of these mycotoxins, are not yet clear [20,33,45,46,47]. Several antioxidant compounds have been studied for their protective effects against OTA-induced organ toxicity [30,46,48]. However, there is no study about the potential effects of CURC on OTA-induced hepatotoxicity in rats. The protective effects of CURC on OTA-induced liver oxidative injury have been recently demonstrated in ducks, where CURC was able to restore the liver CAT activity with good recovery of the OTA-induced alteration in the composition of intestinal microbiota [25]. Therefore, the present work has been conducted to investigate, for the first time, the possible hepatoprotective effects of CURC against OTA intoxication in rats. Although the liver is not the main target organ for OTA, it remains one of the most sensitive organs to toxic exogenous substances as hepatocytes are exposed to OTA, which must pass through the liver after intestinal absorption [49]. Additionally, OTA was suggested to cause an imbalance between oxidant/antioxidant parameters in both the kidney and liver of rats [50]. Considering that some polyphenolic compounds with established antioxidant properties may be used in the prevention of the induced hepatotoxicity [51], the current study focused on the antioxidant effects of CURC in reducing OTA oxidative stress in the liver of rats. Considering the chemical properties of CURC such as low water solubility, poor stability in body fluids, high rate of metabolism, rapid clearance, reduced absorption in the gastrointestinal tract, and limited bioavailability [52,53], numerous innovative approaches have been employed in recent years to increase its solubility and bioavailability [54,55]. To increase the stability and intestinal absorption of CURC, olive oil was used as a solvent, according to other data in the literature [56]. The liver is constantly negatively affected by various xenobiotics and free radicals, as it is the organ in charge of biotransformation processes. In fact, several studies reported CURC as a scavenger for the oxygen-derived free radicals [34,57]. In this study, we analyzed the main serum biomarkers essential for the determination of liver injury [58,59], and we found, in rats treated with OTA for 14 days, a significant increase in serum aminotransferases (ALT and AST) and ALP that suggests damage to the liver cells. Moreover, the total protein concentrations were significantly reduced, and this may have been due to loss of hepatocytes as a result of OTA damage, leading to disturbances of protein biosynthesis. In contrast, cotreatment with CURC significantly reduced the serum levels of ALT, AST, and ALP, evidencing the protective effects of CURC. CURC has been shown to prevent the destruction of liver enzymes in the serum and thus protect the hepatocytes from OTA-induced hepatotoxicity. The significant reduction in the activities of these enzymes in the curcumin-treated group suggests a chelating effect of curcumin. A significant increase in the serum total protein concentration was observed in the group treated with CURC administrated in association with OTA when compared to the OTA group. This study, according to the studies of Emad et al. [59] and Omowumi et al. [60], allows us to hypothesize that CURC increases protein synthesis in liver cells damaged by OTA for a regenerative effect on liver tissue. Mycotoxins cause the release of free radicals, induce lipid peroxidation, and change the antioxidant status of cells, leading to oxidative stress [34,61]. Therefore, in this study, we have evaluated the activity of the main antioxidant enzymes, namely CAT, SOD, and GPx, which play a key role in protecting against oxidative stress induced by reactive oxygen species [62]. Here, the activities of CAT, SOD, and GPx were significantly reduced in the liver of OTA-treated rats; according to data in the literature [63], mainly the activity of GPx and selenoenzyme catalyzes the reduction of hydrogen peroxide to water utilizing glutathione (GSH), which can be attributed to the decrease in selenium levels due to OTA interference with its essential uptake. Therefore, according to the studies of Palabiyik et al. [50] and Domijan et al. [64], we hypothesize that the toxic action of OTA is partly related to the reduction in the antioxidant enzyme activities and partly to the increase in free radical generation. In fact, the high generation of free radicals causes an imbalance between oxidant and antioxidant systems inducing oxidative stress. In accordance with the data in the literature [19,65] OTA causes the overproduction of free radicals determining damage to cell constituents, such as membrane lipids. In the current study, after 14 days of treatment, an increase in the MDA level in the liver tissue was observed in the OTA-treated rats. The increase in MDA levels can be considered as a marker of tissue injury induced by OTA. The literature reports that the beneficial effects of CURC are linked to its antioxidant properties, which play a key role in hepatoprotective mechanisms, through the elimination of free radicals and strengthening of the activity of the natural antioxidant defense system [66,67,68,69,70]. In this study, the treatment with CURC, used alone or in association with OTA, revealed a significant reversion of the activities of CAT, SOD, and GPx and MDA levels, inhibiting effects of oxidative stress induced by OTA. 

The antistress activity of CURC may be partly due to its antioxidant effect, which leads to an elimination of free radicals and a decline in the biosynthesis of mycotoxins [71], demonstrating potent antifungal and antimycotoxin activities on *A. ochraceous* and *P. verrucosum* [72]. 

These findings were supported by liver histopathological analysis, where the multifocal lymphoplasmacellular hepatitis, periportal fibrosis, and hepatocellular necrosis observed in the OTA group were consistent with the lesions already reported in the livers of rats treated with OTA [40]. Central vein dilation in the OTA group was consistent with the data obtained by previous studies [42] and might be due to developing hypertension secondary to the OTA-dependent renal damage [45]. The less severe inflammation, steatosis, and necrosis scores reported in the rats of the OTA + CURC group compared to the OTA group are consistent with the molecular results.

## 5. Conclusions

Our results confirmed that oxidative stress is involved in the OTA hepatotoxicity mechanism and exposure to OTA induces negative effects and profound changes in liver functions. CURC treatment was responsible for a restoration in liver function, biochemical and antioxidant enzyme activities, and MDA levels. Therefore, the use of CURC, due to its antioxidant activity, could overcome oxidative stress and decrease the biosynthesis of mycotoxins in food sources, while protecting human and animal health.

## Figures and Tables

**Figure 1 antioxidants-10-00125-f001:**
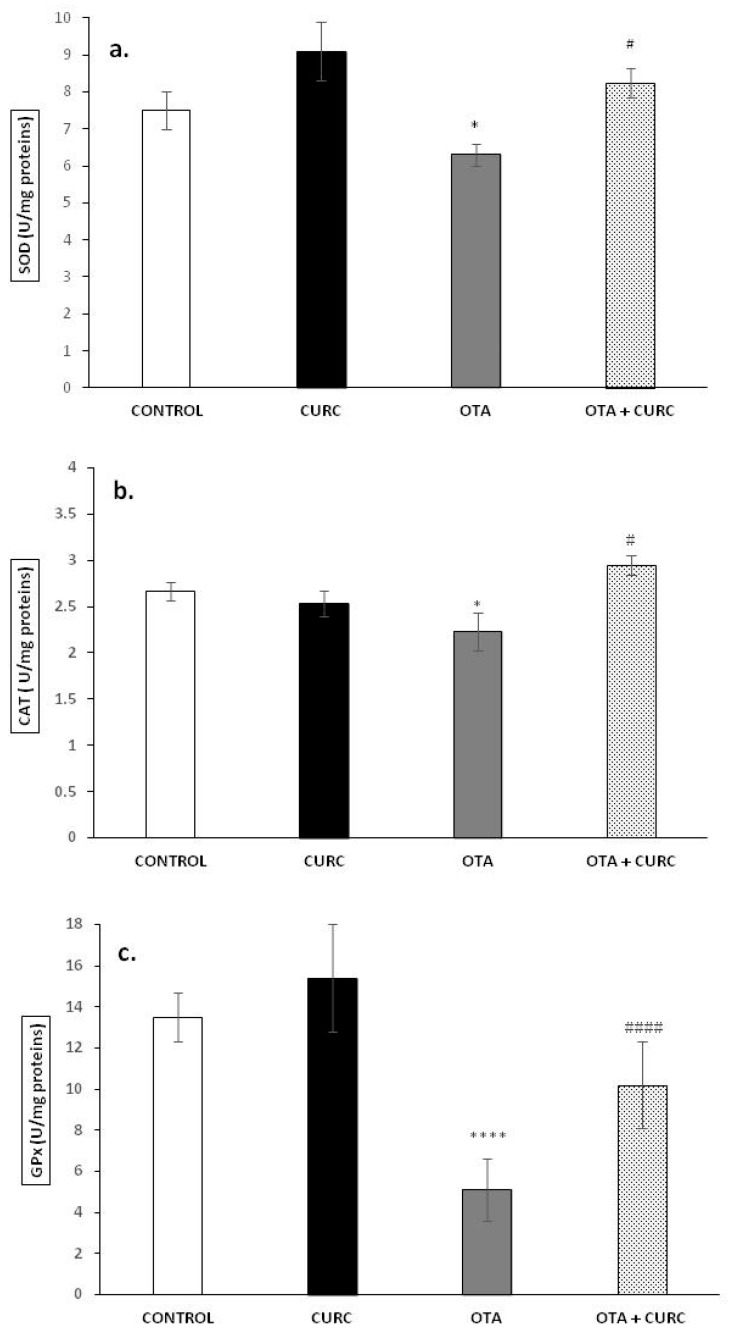
Effects of curcumin (CURC) on superoxide dismutase (SOD), catalase (CAT), and glutathione peroxidase (GPx) activities expressed as units per milligram of protein (U/mg proteins) in liver tissue of experimental groups after 14 days of treatment. (**a**) Liver SOD activity; (**b**) liver CAT activity; (**c**) liver GPx activity. Control group (CONTROL); curcumin group (CURC); ochratoxin A group (OTA); curcumin plus ochratoxin A group (CURC + OTA). Data are expressed as mean ± standard deviation (SD) of *n* = 6 rats. OTA treatment significantly decreased SOD, CAT, and GPx enzyme activities, while coadministration with CURC significantly restored this effect (* *p* < 0.05 and **** *p* < 0.0001 vs. CONTROL; ^#^
*p* < 0.05 and ^####^
*p* < 0.0001 vs. OTA).

**Figure 2 antioxidants-10-00125-f002:**
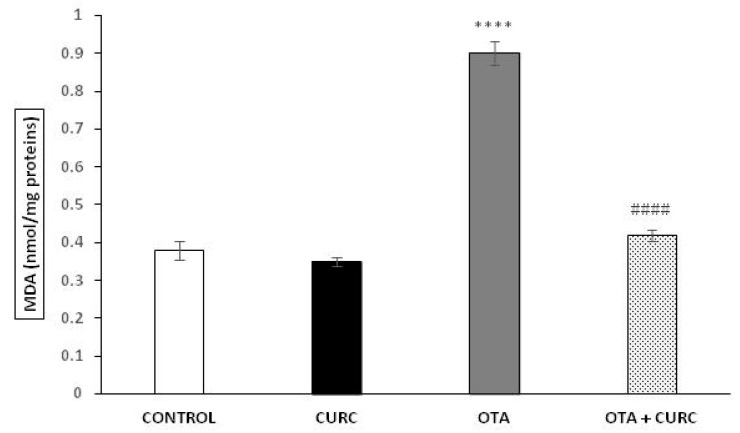
Effect of curcumin (CURC) used alone or in association with ochratoxin A (OTA) on lipid peroxidation measured by malondialdehyde (MDA) expressed in nanomoles of MDA per milligram of protein (nmol/mg proteins) in rat liver after 14 days of treatment. CONTROL group (CONTROL); curcumin group (CURC); ochratoxin A group (OTA); ochratoxin A plus curcumin group (OTA + CURC). Results are expressed as mean ± standard deviation (SD) of *n* = 6 rats. OTA treatment significantly increased MDA levels, while coadministration with CURC prevented this effect (**** *p* < 0.0001 vs. CONTROL; ^####^
*p* < 0.0001 vs. OTA).

**Figure 3 antioxidants-10-00125-f003:**
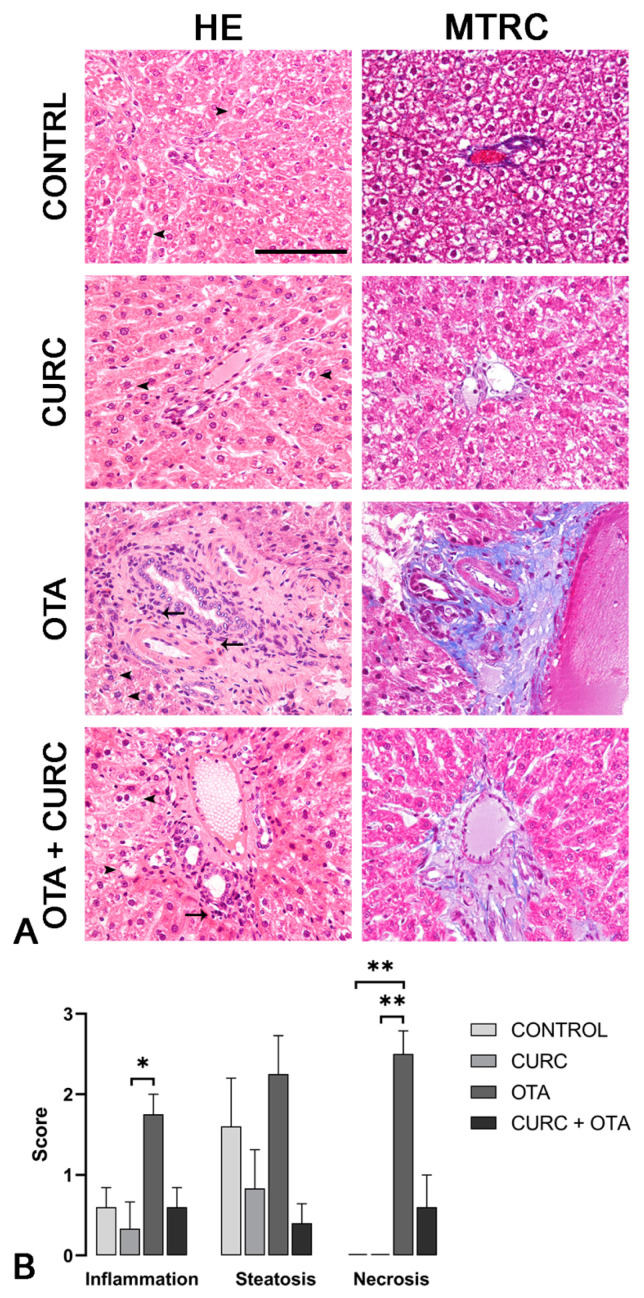
(**A**) Rats, liver, hematoxylin and eosin (HE) and Masson’s trichrome (MTRC) stains, 40× magnification, scale bar = 100 µm. Control group (CONTROL); curcumin group (CURC); ochratoxin A group (OTA); ochratoxin A plus curcumin group (OTA + CURC). (**A**) Rats of the CONTROL and CURC groups showed numerous disseminated swollen hepatocytes with intracytoplasmic optically empty vacuoles (steatosis, arrowheads) with HE stain and a normal amount of interstitial connective tissue (blue) with MTRC stain. OTA-treated rats showed periportal infiltration of lymphocytes (arrows) and numerous disseminated swollen hepatocytes with intracytoplasmic optically empty vacuoles (steatosis, arrowheads). Rats of the OTA group also showed portal spaces severely expanded by abundant fibrous connective tissue (blue) with MTRC stain. Rats of the OTA + CURC group showed focal periportal infiltration of lymphocytes (arrow) and numerous disseminated swollen hepatocytes with intracytoplasmic optically empty vacuoles (steatosis, arrowheads) with HE stain. Rats of the OTA + CURC group showed portal spaces moderately expanded by fibrous connective tissue (blue) with MTRC stain. (**B**) Severity scores of inflammation, steatosis, and necrosis for each group. Asterisks represent statistically significant differences between groups (* *p* < 0.05, ** *p* < 0.01).

**Table 1 antioxidants-10-00125-t001:** Serum biochemical parameters: alanine aminotransferase (ALT), aspartate aminotransferase (AST), and alkaline phosphatase (ALP) activities expressed in units per liter (U/L) and total protein expressed in grams per deciliter (g/dL) in the different groups of rats after 14 days of treatment. Control group (CONTROL); curcumin group (CURC); ochratoxin A group (OTA); ochratoxin A plus curcumin group (OTA + CURC). Data are expressed as mean ± standard deviation (SD) of *n* = 6 rats. (* *p* < 0.05 and **** *p* < 0.0001 vs. CONTROL; ^#^
*p* < 0.05, ^###^
*p* < 0.001, and ^####^
*p* < 0.0001 vs. OTA).

Groups	ALT (U/L)	AST (U/L)	ALP (U/L)	TOTAL PROTEIN (g/dL)
CONTROL	45.33 ± 6.4	118.82 ± 1.45	25.60 ± 5.8	81.55 ± 3.49
CURC	40.11 ± 5.3	100.50 ± 6.2	21.50 ± 5.6	84.47 ± 1.35
OTA	102.21 ± 5.8 ****	188.91 ± 4.6 ****	88.32 ± 4.9 ****	53.25 ± 2.55 *
OTA + CURC	78.20 ± 4.7 ^####^	157.72 ± 6.1 ^###^	57.31 ± 5.2 ^###^	77.14 ± 2.36 ^#^

## Data Availability

The data sets used and/or analyzed during the current study are available from the corresponding authors.

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
