# Peer review of "Antioxidative Effects of Curcumin on the Hepatotoxicity Induced by Ochratoxin A in Rats"

_antioxidants, 2021, doi:10.3390/antiox10010125_

Round 1
Reviewer 1 Report
Antioxidant effect of curcumin on the liver toxicity induced by mycotoxin Ochratoxin A in rat models.
It is well known that curcumin has many potential beneficial effects including antioxidant effect. Mycotoxins cause oxidative stress. Oxidative stress also causes chronic kidney disease, hepatic
inflammation, hypercholesterolemia, diabetes, hepatic cirrhosis. So, curcumin role in mycotoxin Ochratoxin A may be very useful.
In experimental design author used vehicle control group and OTA control group with 2ml/kg bw olive oil while Curc and Curc + OTA group having 1ml/kg bw olive oil. Reduced consumption of olive oil in different groups? Why such changes required?
Liver SOD and catalase levels are mild change? Author may need to show other strong readout ?.
Figure 3A: What is the scale in H&E stained image? H&E images are not clearly visible in inflammation and injury? Need to show higher magnified images?
Figure 3B: levels need to rearrange? Statistical significance not properly showed? In the figure, there is no significance showed in OTV vs CURC +OTA??
This study is not unique of curcumin against OTA toxicity. Curcumin against OTA nephrotoxicity and liver toxicity has been reported (author need to cite those articles and describe all other relevant reported finding). PMID: 32036964, PMID: 32325727
Author Response
Response to Reviewer 1 Comments
Dear Reviewer,
Me and my co-authors are grateful for your useful contribution that helped us to improve the quality of our manuscript.
We believe that we addressed all the criticisms raised, in the manuscript and listed below with point-by-point reply:
Point 1: In experimental design author used vehicle control group and OTA control group with 2ml/kg bw olive oil while Curc and Curc + OTA group having 1ml/kg bw olive oil. Reduced consumption of olive oil in different groups? Why such changes required?
Response 1: Many apologies for the mistake. The olive oil consumption in Control group and Curc + OTA was the same. The phrase was corrected according referee’s suggestion
Point 2: Liver SOD and catalase levels are mild change? Author may need to show other strong readout ?.
Response 2:Liver SOD, CAT and GPX are significant decresed in OTA treated group according with data in literature (Zugong, Y et al Mol Med Rep. 2018, 18(3), 2551–2560). In particular we found P<0.05 in SOD and CAT activity and P<0.0001 in GPx activity and co-treatment with CURC restored these values. Moreover, also Zhai S.S and colleagues have found a significant reduction in CAT and a trend in reduction of SOD and GSH activity in OTA-treated duck and a significant protective effect of curcumin on CAT activity was reported by the authors. It is possible that the small difference observed by us is related to the difference in species (rat and duck), in the OTA treatment (our treatment is by OTA Sigma- Aldrich by gavage for 14 days, while Zhai et al used an OTA-contaminated corn for 3 week), as well as the kit used.
Point 3: Figure 3A: What is the scale in H&E stained image? H&E images are not clearly visible in inflammation and injury? Need to show higher magnified images?
Response 3: We thank the reviewer for this point. A scale bar has been added for the histologic images. In the revised manuscript images has been changed with higher magnified images (40x magnification).
Point 4: Figure 3B: levels need to rearrange? Statistical significance not properly showed? In the figure, there is no significance showed in OTV vs CURC +OTA??
Response 4: We thank the reviewer for this point. The graph position has been rearranged. The significance levels shown have been double checked. As described in the results there are no significative differences between OTA and CURC + OTA groups.
Point 5: This study is not unique of curcumin against OTA toxicity. Curcumin against OTA nephrotoxicity and liver toxicity has been reported (author need to cite those articles and describe all other relevant reported finding). PMID: 32036964, PMID: 32325727
Response 5: Many thanks for your comment. Although our sentence in the main text of the discussions is referred to curcumin against OTA liver toxicity in the "rat" and no bibliographic references are found for this species, we have modified the text by adding liver toxicity to a different species such as the "duck", citing reference as kindly suggested by you.

Reviewer 2 Report
This study was aimed to evaluate anti-oxidative effects of curcumin on the hepatotoxicity induced by Ochratoxin A in rats. However, recent study have found that curcumin shows great inhibitory effect on Ochratoxin A–induced liver oxidative injury (Poultry Science, 2020 99:1124–1134). The authors should address the novelty of present study in the abstract and introduction, and discuss the present findings with the previous study (Poultry Science, 2020 99:1124–1134). Also, a similar study conducted by your research group has been found to be published in this journal (Antioxidants 2020, 9, 332). Were the data collected in the present study from the same animal experiment as previously reported? The author should explain this issue in the Materials and Method.
Author Response
Response to Reviewer 2 Comments
Dear Reviewer,
Me and my co-authors are grateful for your useful contribution that helped us to improve the quality of our manuscript.
We believe that we addressed all the criticisms raised, in the manuscript and listed below with point-by-point reply:
Point 1 : This study was aimed to evaluate anti-oxidative effects of curcumin on the hepatotoxicity induced by Ochratoxin A in rats. However, recent study have found that curcumin shows great inhibitory effect on Ochratoxin A–induced liver oxidative injury (Poultry Science, 2020 99:1124–1134). The authors should address the novelty of present study in the abstract and introduction, and discuss the present findings with the previous study (Poultry Science, 2020 99:1124–1134). Also, a similar study conducted by your research group has been found to be published in this journal (Antioxidants 2020, 9, 332). Were the data collected in the present study from the same animal experiment as previously reported? The author should explain this issue in the Materials and Method.
Response 1: Many thanks for your comment. Although our sentence in the main text of the discussions referred to liver toxicity of OTA and the protective effect of curcumin is related to the "rat" and no bibliographic references are found for this species, we have modified the text by adding liver toxicity to a different species such as the "duck", citing reference as kindly suggested by you.
About the data collected in the present study, the animal experiment groups are the same of previously reported paper. We have explained this issue in the Materials and Method. Thanks for the suggestion.

Round 2
Reviewer 1 Report
There is no behavioral data found in the manuscript. In addition, no description of behavioral study related discussion in manuscript.
Author Response
Dear Review
thanks for your support.
In this experimental work, we didn't evaluated the behavioral data
Regards
Roberto Ciarcia
Reviewer 2 Report
The authors have added ref as indicated in previous comment; however, the novelty was not found to be mentioned in the introduction and abstract. If the species of animal may be a factor that lead to different responses of curcumin against hepatotoxicity, the author should clearly describe this point as the rationale of present study in the introduction and abstract, as well as collecting more ref to support this. Besides, sentence of Line 333-336 addressed "This data was confirmed in our previous work where we found that the CURC increased the body weight of rats poisoned by OTA after 14 days of treatment , recalling that the body weight of an animal is a main indicator of the toxic impacts of various drugs or chemicals on animals." is inappropriate, because the study cited and present study are the same research work. The author should check this issue throughout the manuscript to avoid the violation of academic ethics.
Author Response
Dear Reviewer
as indicated by the guidelines of the Antioxidants journal, it is not possible to add “results wich are not presented and substantianted in the main text and should not exaggerate the main conclusion”. However, we had already expanded the information to species other than rats, as indicated in line 28-29. Moreover, the novelty that you're referring about the duck is added in line 103-106 also in the introduction.
Untilnow, the OTA response to difference in species was far from clear (Yanfei Tao et al, food and Chemical Toxicology 112 (2018) 320-331). Various in vivo and in vitro studies, have identified that the induction of an oxidative stress response induced by OTA exposure is considered one of the mechanisms responsible for its toxicity but several diverse metabolites of OTA was found in different species, as such as rats,rabbits, goats, pigs and humans. May be the metabolic differences of each species, including the rat and the duck, are the cause of the slight differences found, according to several data in literature (Yang et al,2015; Pinelli et al 1999; Hohler et al,1976). We have reported in the main test, according referee’s suggestion (line 107-110)
To avoid the violation of academic ethics, we have deleted the sentence and our reference, but added the references of other authors who have used olive oil to keep the bioavailability of CURC, works that we have used to plan our experiments. The sentence “body weight of an animal is a main indicator of the toxic impacts of various drugs or chemicals” was not ours sentence, but it is clear indicator of toxic impacts of various drugs, as well as indicator by Wang, C et all (PloS ONE 2016). However, the phrase was deleted according refereee’s suggestion (line 336-340).
Regards
Roberto Ciarcia
Round 3
Reviewer 2 Report
The author has made a detailed revision according to the reviewer's suggestion. I have no more comments.